# Multiresponse Optimization of Selective Laser Melting Parameters for the Ni-Cr-Al-Ti-Based Superalloy Using Gray Relational Analysis

**DOI:** 10.3390/ma16052088

**Published:** 2023-03-03

**Authors:** Anton V. Agapovichev, Alexander I. Khaimovich, Vitaliy G. Smelov, Viktoriya V. Kokareva, Evgeny V. Zemlyakov, Konstantin D. Babkin, Anton Y. Kovchik

**Affiliations:** 1Engine Production Technology Department, Samara National Research University, 34 Moskovskoye Shosse, 443086 Samara, Russia; 2Turbomachinery and Heat Transfer Laboratory, Aerospace Engineering Department, Technion-Israel Institute of Technology, Haifa 3200003, Israel; 3World-Class Research Center “Advanced Digital Technologies”, State Marine Technical University, 190121 Saint Petersburg, Russia

**Keywords:** additive manufacturing, selective laser melting, Ni-Cr-Al-Ti, metal powder, mechanical properties, microstructure, gray relational analysis

## Abstract

The selective laser melting technology is of great interest in the aerospace industry since it allows the implementation of more complex part geometries compared to the traditional technologies. This paper presents the results of studies to determine the optimal technological parameters for scanning a Ni-Cr-Al-Ti-based superalloy. However, due to a large number of factors affecting the quality of the parts obtained by selective laser melting technology, the optimization of the technological parameters of the scanning is a difficult task. In this work, the authors made an attempt to optimize the technological scanning parameters which will simultaneously correspond to the maximum values of the mechanical properties (“More is better”) and the minimum values of the dimensions of the microstructure defect (“Less is better”). Gray relational analysis was used to find the optimal technological parameters for scanning. Then, the resulting solutions were compared. As a result of the optimization of the technological parameters of the scanning by the gray relational analysis method, it was found that the maximum values of the mechanical properties were achieved simultaneously with the minimum values of the dimensions of a microstructure defect, at a laser power of 250 W and a scanning speed of 1200 mm/s. The authors present the results of the short-term mechanical tests for the uniaxial tension of the cylindrical samples at room temperature.

## 1. Introduction

The current pace of the aviation and rocket and space industry development requires the use of production technologies that make it possible to obtain high-quality products in the shortest possible time at the lowest cost and with minimal post-processing [1,2]. The use of additive manufacturing (AM) can significantly reduce the duration of the technological preparation for the production of new products with a rather complex shape, use fundamentally new design and technological solutions, and ultimately reduce the labor intensity and the cost of manufacturing the products [3]. AM is developing at a faster pace and, above all, processes involving 3D printing of metals and alloys, because they allow the manufacture of geometrically complex components that cannot be easily formed by other traditional processes, and, in addition, minimize the loss of raw materials [4,5].

Ni-based superalloys are the most extensively investigated superalloys in the AM field and have become the materials of choice for fabricating the components destined for high-temperature industrial applications, such as turbine discs and blades [6]. The most commonly processed AM Ni-based superalloys are Inconel 718 and Inconel 625, fabricated with a relative density close to 100% due to their good weldability [7].

Nevertheless, Inconel 718 and Inconel 625 can be used up to around 650 °C for applications under a load. In fact, it is well known that the operative temperature of these alloys is limited under a high load due to the coarsening and the transformation of the metastable γ” phase into the δ phase, which can significantly reduce the mechanical characteristics [8]. In this regard, the need for alloys operating at temperatures above 650 °C causes increased interest in the development of nickel-based alloys by the AM. Hastelloy X, Inconel 738LC and CM247LC are among these superalloys [7]. As recently reported, a high concentration of Si and C may play a critical role in increasing the cracking susceptibility of Hastel-loy X [9]. The ‘Weldability’ diagram for a range of nickel-based superalloys (Figure 1) shows that the CM247LC and Inconel 738LC alloys have been considered poor [10].

Here, the AM of a Ni-20.5Cr-6Al-3.5Ti (wt%) model alloy is studied. This provides a model Ni-base superalloy with a high Al + Ti content and a high γ’ fraction and solvus temperature. High creep resistance, corrosion, oxidation resistance, as well as microstructure stability up to around 850 °C are presented by Ni-Cr-Al-Ti superalloys [11].

Selective laser melting (SLM) is one of the emerging AM technologies that use high-power lasers to create three-dimensional physical objects by fusing metal powders [12,13]. The quality of parts manufactured by SLM technology is affected by a large number of technological parameters [14,15]. By properly understanding and managing these parameters, it is possible to obtain parts of a quality that is not inferior to that of the parts obtained by the traditional production methods. The complexity of the SLM process lies in many thermal, physical, and mechanical interactions and the influence of a large number of technological parameters on them. For example, the SLM Solutions build processor used on the SLM 280HL contains over 100 configurable parameters. [15,16]. There are four groups of technological parameters: laser parameters, scan options, material parameters, and atmospheric parameters [17].

From the literature sources [18,19], it is known that the main technological parameters of scanning are laser power P, (W); scanning speed V, (mm/s); hatch distance h, (µm); layer thickness t, (µm) and scanning strategy type [15]. The technological scanning parameters are often combined into one parameter, which is called volume energy density (VED, J/mm^3^) and is determined by Equation (1). This parameter has no physical meaning, but it is used to compare the physical and mechanical properties of a material synthesized at different technological parameters [20].
(1)$$VED=\frac{P}{V\cdot h\cdot t},$$
where *P* is the laser power, W; *V* is the scanning speed, mm/s; *h* is the hatch distance, mm; and *t* is the layer thickness.

In some papers, the technological parameters of scanning, as a result of which a material with a density close to 100% is synthesized, are called optimal technological parameters [21,22]. To achieve a material density close to 100%, it is necessary to ensure that sufficient heat is supplied to a certain volume of material in order to ensure its complete penetration. As can be seen in Equation (1), this condition can be met by applying various combinations of technological scanning parameters. The numerous combinations of technological scanning parameters, as a result of which a material with a density close to 100% is synthesized, are called the range of optimal technological parameters of scanning (process window). For example, Gong et al. determined the process window for the Ti-6Al-4V titanium alloy [23]. The process window is divided into four areas, and a material with a density close to 100% can be synthesized using one of the combinations of the laser power and the scanning speed. In the work of Vrancken B. [24] on the process window for the SLM of AlSi10Mg and in the work of Montgomery C. et al. [25] on the process window for the SLM of Ti6Al4V studies of the data were carried out on the shape of the melt pools formed as a result of exposure to technological scanning parameters.

However, the quality of the material synthesized by SLM technology is characterized by more than one experiment response characteristic, which can weakly correlate with each other [1]. Therefore, when we faced the problem of simultaneously optimizing several quality characteristics (yield strength, tensile strength, relative elongation, area of non-melting points, pore diameter, etc.), it became clear that it was necessary to find a compromise between their optimal states, depending on the importance of one quality parameter compared to the others.

Gray relational analysis (GRA) is a popular type of optimization method which is used for solving of multi-criteria problems [26,27]. In gray relational analysis, the dimensions of the factors considered are usually different, and their magnitude difference is large. Therefore, the original data are normalized to make the magnitude of the original data of the order of one and to make them dimensionless [28]. Therefore, in order to search for the optimal technological parameters of scanning, the initial experimental data are first normalized in the range from 0 to 1 in accordance with the principles “More is better” for the values of the mechanical properties (target value 1) and “Less is better” for the sizes of the microstructure defects (target value 0). The overall evaluation of the multiple quality characteristics is based on the grey relational grade [29]. Firstly, the gray relational coefficient is determined to quantify the relationship between ideal and actual experimental results. Then, to summarize all the quality characteristics, the grey relational grade is calculated by averaging the gray relational coefficients. The optimal level of the process parameters is the level with the highest grey relational grade [26,28].

## 2. Materials and Methods

In this paper, a study to determine the optimal technological scanning parameters for the Ni-Cr-Al-Ti-based superalloy to be processed by SLM was conducted based on the gray relational analysis. In the gray relational analysis, the raw experimental data are normalized at first for the reasons discussed earlier. In this study, a linear normalization of the experimental results is performed in the range of 0 to 1, which is called the gray relational generating.

The normalized results for the “more is better” quality characteristic can be expressed as:(2)$$x_{ij}=\frac{y_{ij}-\min_{j}y_{ij}}{\max_{j}y_{ij}-\min_{j}y_{ij}},$$

For the “less is better“ quality characteristic, the normalized results can be expressed as:(3)$$x_{ij}=\frac{\max_{j}y_{ij}-y_{ij}}{\max_{j}y_{ij}-\min_{j}y_{ij}},$$
where $y_{ij}$ is the value of the quality parameter *j* for the *i*-th combination of the scanning parameters, and $y_{ij}=\max_{n}y_{ij}^{n}$ is the most negative quality characteristic among the *n =* 1, …, 9 combinations of scanning parameters.

The gray relational coefficients $\xi_{ij}$, which are calculated to determine the relationship between the ideal and the actual results of an experiment, can be expressed as [28,29,30]:(4)$$\xi_{ij}=\frac{\min_{i}\min_{j}|x_{j}^{0}-x_{ij}|+\zeta \max_{i}\max_{j}|x_{j}^{0}-x_{ij}|}{\left|x_{j}^{0}-x_{ij}|+\zeta \max_{i}\max_{j}|x_{j}^{0}-x_{ij}\right|},$$
where $x_{i}^{0}$ is the ideal normalized result (i.e., best normalized result) for the *j*-th quality characteristics, and ζ = [0,1] is a distinguishing coefficient, the purpose of which is to weaken the effect of $\max_{i}\max_{j}|x_{j}^{0}-x_{ij}|$ when it gets too big and thus enlarges the difference significance of the relational coefficient. In general, it is set to 0.5 if all the process parameters have equal weighting [29].

Based on the experience of using gray relational analysis to search for rational SLM process parameters, the recommended values of the coefficient ζ are given in Table 1. The same table shows the normalized desired quality characteristics ($x_{i}^{0}$).

After obtaining the gray relational coefficients, a weighting method is used to integrate the gray relational coefficients of each experiment into the gray relational grade. The quality characteristics are based on the gray relational grade and can be expressed as [29,30]:(5)$$\gamma_{i}=\frac{1}{m}\sum_{j=1}^{m}\xi_{ij},$$
where *m* is the number of quality characteristics.

In calculating the gray relational grades, the weighting ratio for the quality characteristics are set as 1:1, i.e., each characteristic has equal importance or relative weighting.

The Ni-Cr-Al-Ti-based superalloy metal powder was used as an initial material. Due to the scanning with the electron microscope Tescan Vega and Energy Dispersive X-rays Spectrometer INCAx-act it was possible to investigate the morphology of surface powder and chemical composition.

The following factors were chosen as variable factors: laser power (P), scanning speed (V), and hatch distance (h). The research by Anthony De Luca et al. shows that the highest density of the Ni-Cr-Al-Ti superalloy achieved is achieved at a VED of 66.6 J/mm^3^, and the lowest is at a VED of 44 J/mm^3^ [31]. However, a publication by Gokhan Dursun et al. states that there is no relationship between the material density and VED [32]. Therefore, the factors varied in the following ranges (process window from min. to max.): laser power 250, …, 350 W, scanning speed 600, …, 1200 mm/s, and hatch distance 0.09, …, 0.15 mm, which correspond to a VED of 34.7, …, 97.2 J/mm^3^. The samples were prepared at a layer thickness of 0.05 mm; the scanning strategy was zigzag.

Samples were fabricated from Ni-Cr-Al-Ti based superalloy powder on a SLM 280HL 3D printer (SLM Solutions Group AG, Lubeck, Germany). This 3D printer operates using SLM technology, the essence of which is laying the powder manufacturing a part by fusing layers together [6].

The optimal scanning parameters determination was implemented on proportional flat samples with dimensions (L × W × H) 70 × 2 × 15 mm. The study was accomplished ac-cording to the fractional factorial design 3 (3-1) with 3 times repetition for each experiment. The experimental design is demonstrated in Table 2.

The mechanical test of the sintered specimens was realized on Instron 8802.

To appoint the non-melting area and the diameter of the pores, thin sections of the cross section of the samples were arranged. Etching of the samples was achieved by electrolytic method at room temperature for 5...10 s in an electrolyte of the following composition: 10 g of citric acid + 10 g of ammonium chloride + 1 L of water.

Microanalysis was implemented on an optical microscope METAM LV-41 in a bright field with ×50...200 magnification. The processing of the acquired images of the microstructure was accomplished in a specialized software product SIAMS.

## 3. Results and Discussion

### 3.1. Powder Distribution

The general view of the powder particles is shown in Figure 2. The electron microscope analysis showed that the powder particles mainly have a spherical shape (92%), which is typical for the method of obtaining powders by melt dispersion [33]. The particle size varies in the range of 15–45 microns. The plus and minus fractions are 0.5% and 1.5%, respectively.

The microspectral analysis of the powder particles showed that the chemical composition of the Ni-Cr-Al-Ti-based superalloy powder complies with the manufacturer’s quality certificate (TU 1809104096), except for the Al content. The chemical composition of the powder is presented in Table 3.

### 3.2. Microstructural Characterization and Mechanical Property Tests

During macroanalysis, it was found that on the surface of all the samples there were pores less than 0.1 mm in size. Large defects (non-melting area) of up to 0.6 mm in size were found on sample Nos. 7 and 9 (Figure 3).

In the central part of all the samples, except for sample Nos. 7 and 9, there were single pores (up to 0.03 mm) and non-melting area with a maximum size of 0.1 × 0.25 mm (Figure 4). In the structure of sample Nos. 7 and 9, multiple non-melting spots of up to 0.3 × 0.6 mm in size were observed.

According to the literature, the metal melting is classified into two modes: the conduction mode and the mode with the formation of a penetration channel (melting with deep penetration) [34]. According to the results of the studies by the authors W.E. King and others [35] and E.W. Eagle and others [36], the cross-section of the melt pool formed in the conduction mode is approximately semicircular. Figure 5 shows that the use of the modes with a VED from 34.7 to 66.7 J/mm^3^ leads to the formation of a melt pool which is close to semicircular. An increase in VED leads to the transition of the melting of the material to the formation of a penetration channel. An increase in VED to 97.2 J/mm^3^ leads to the melting of the material in the deep penetration mode and the formation of keyhole defects. For further analysis and the establishment of a correlation between the technological parameters and the characteristics of the structure, it was necessary to convert the descriptive characteristics of the microstructure into quantitative ones.

For this purpose, the rank of the desirability of the characteristics of the structure of the resulting material was assessed based on the data in Table 4.

In comparison to the as-casted samples, Table 5 reveals that the SLM-manufactured Ni-Cr-Al-Ti superalloy had high tensile strength, much higher than that of the as-casted samples (768 MPa [37]). The highest tensile strength achieved was 949 MPa (300 W and 600 mm/s, equivalent to 66.7 J/mm^3^), and the lowest was 774.7 MPa (250 W and 900 mm/s, equivalent to 37 J/mm^3^). The SLM-manufactured Ni-Cr-Al-Ti superalloy had high relative elongation, much higher than that of the as-casted samples (3.3 % [37]). The highest relative elongation achieved was 10.2% (250 W and 1200 mm/s, equivalent to 34.7 J/mm^3^), and the lowest was 5.2% (250 W and 900 mm/s, equivalent to 37 J/mm^3^)

The SLM-manufactured Ni-Cr-Al-Ti superalloy had yield stress elongation, slightly less than that of the as-casted samples (703 MPa % [37]), on samples 2, 3, 6, 7, and 9. In general, the data presented in Table 5 indicate a large variability of the process, which requires the use of additional analysis to select the optimal fusion mode.

To determine the method for finding the optimal scanning parameters, a correlation analysis of the experimental results was carried out; it determined the statistical significance of the relationship between the technological parameters (factors) and each response of the experiment (quality characteristic). If the relationship is significant then Spearman’s correlation coefficient will be greater than 0.7, otherwise the relationship is considered insignificant. In the case of a significant relationship between the factors and the experiment responses, it is possible to establish a regression dependence, which will have sufficient reliability in terms of the totality of the characteristics, including the coefficient of determination and Fisher’s F-test.

If the correlation coefficients do not confirm the statistical significance of the relationships between the factors and the responses (either all or some of them), then the gray relational analysis method can be used to select the optimal scanning parameters among the nine experiments performed.

The correlation matrix of the factors and characteristics of the experiment response, as presented in Table 6, shows that the range of changes in the scanning parameters in the coefficients of the correlation matrix is not statistically significant, with the exception of the relationship between the pore diameters, the energy density, and the scanning speed. In addition, the mechanical properties have a weak correlation with the rank assessment of the microstructure characteristics. Thus, in agreement with the statement that in the conditions of the significant variability of the process responses, the analysis methods with fuzzy boundaries of target indicators, such as that of gray relational analysis, should be used. So, according to the combination of features, it was advisable to apply the method of gray relational analysis to determine the optimal scanning parameters for the Ni-Cr-Al-Ti-based superalloy powder. An analysis of the influence of VED and scanning speed V on the pore diameter (Table 6) based on the values of the correlation coefficients equal to 0.71002 and −0.77029, respectively, shows that excess fusion energy increases the tendency to form large pores, and a higher speed of movement of the melt pool promotes their reduction. The influence of these two factors has an almost equal effect. Thus, according to the correlation analysis data, it is clear that the optimal technological parameters for the series of experiments performed lie in the region of lower VED values and higher scanning speed values. This result is justified by the fact that the increasing of the fusion energy leads to the formation of solidification cracks, which is related to the keyhole effect (Figure 5—№3) as well as the pores; the latter is due to the evaporation of small powder particles. However, on the other hand, a lack of energy leads to the appearance of non-melting areas (Figure 4—image on the right; Figure 5—№7). For a more detailed analysis and selection of the optimal mode for growing samples, a gray relational analysis was carried out.

### 3.3. Gray Relational Analysis

Gray relational analysis is used to analyze the quality indicators (experiment response characteristics) due to their multi-directionality (different degrees of desirability “more is better” and “less is better”). Table 7 shows the normalized experiment response characteristics obtained using Formulas (1) and (2).

The calculated values of the gray relational coefficient $\xi_{ij}$ obtained using Formula (3) are presented in Table 8. The resulting relational quality scores obtained using Formula (4) are shown in Figure 6. The higher the integrated relational score, the better the result of the experiment, and the closer it is to the ideally normalized value. According to the calculated values, it can be seen that the fourth (0.688) sample has the maximum value of the integral relational assessment. Therefore, the best quality performance is achieved by using a combination of the following optimal scanning parameters: laser power of 250 W and scanning speed of 1200 mm/s, which corresponds to a VED of 34.7 J/mm^3^.

Specimen geometry has to comply with ISO 6892-1:2019 and ASTM E8/E8M-16a standards. The Instron 8802 testing machine was used to evaluate the tensile properties at room temperature, with a displacement rate of a crosshead of 0.0075 mm/s. The uniaxial tensile testing results obtained for the cylindrical samples are summarized in Table 9.

## 4. Conclusions

In this work, gray relational analysis was used to determine the optimal scanning parameters for the Ni-Cr-Al-Ti-based superalloy powder. In order to obtain an overall quality score for multiple quality characteristics of the SLM process, a single one called grey relational grade was used with a simplified optimization procedure. The subsequent analyses and the experiments design technique are carried out to evaluate the optimal parametric combinations. The possibility of the simultaneous optimization of several quality parameters (yield stress, tensile strength, relative elongation, non-melting area, pore diameter, etc.) has been demonstrated.

The analysis of the obtained results made it possible to establish the following:-The level of tensile strength on the test samples varied from 774.7 to 949.0 MPa, with a relative elongation from 5.2% to 10.2%;-The highest tensile strength (949 MPa), with an average level of plasticity (8.4%), was obtained on the samples manufactured using technological modes corresponding to a VED of 66.7 J/mm^3^ at a laser power of 300 W, a scanning speed of 600 mm/s, and a hatch distance of 0.15 mm;-The highest ductility (10.2%), with a high level of tensile strength (908.4 MPa), was observed on the samples manufactured using technological modes corresponding to a VED of 34.7 J/mm^3^ at a laser power of 250 W, a scanning speed of 120 mm/s, and a hatch distance of 0.12 mm.

The correlation analysis results, as applied to the technological regime influence analysis of the sample growth on the large pore formation, showed that the optimal technological parameters for the series of performed experiments belonged to the scope of the lower VED values and the higher scanning speeds. For a more detailed analysis and selection of the optimal mode for the growing samples, taking into account the multiple quality characteristics, a gray relational analysis was carried out.

As a result of the gray relational analysis, it was found that the optimal values of the mechanical properties, microstructure, and defect sizes were achieved by combining the following technological scanning parameters: laser power of 250 W and scanning speed of 1200 mm/s.

The results of the uniaxial tensile tests of the cylindrical samples made using the optimal scanning parameters made it possible to establish that the mechanical properties of the Ni-Cr-Al-Ti-based heat-resistant alloy synthesized by the SLM technology exceeded the mechanical properties of the same material obtained by the casting technology.

So, the average yield strength of the heat-resistant material obtained by the SLM technology was 621.2 MPa, which was 11,6% less than the standard values for this alloy. The average tensile strength of the heat-resistant material obtained by the SLM technology was 798.7 MPa, which was 3,8% higher than the standard values for this alloy. The average relative elongation of the heat-resistant material obtained by the SLM technology was 20.4%, which was 83.8% higher than the standard values for this alloy.

## Figures and Tables

**Figure 1 materials-16-02088-f001:**
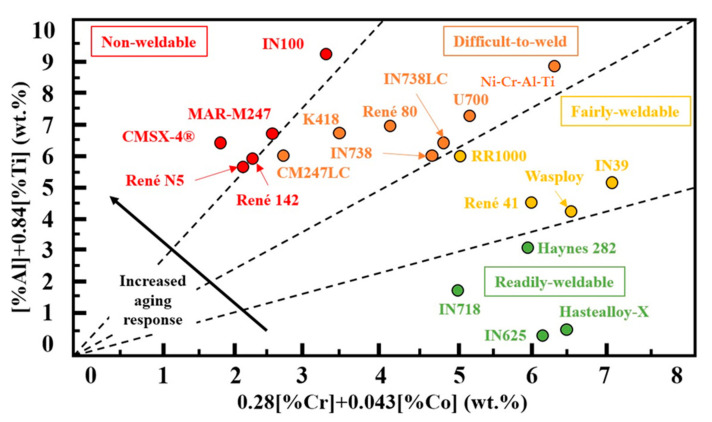
Weldability assessment for nickel-based superalloys [10].

**Figure 2 materials-16-02088-f002:**
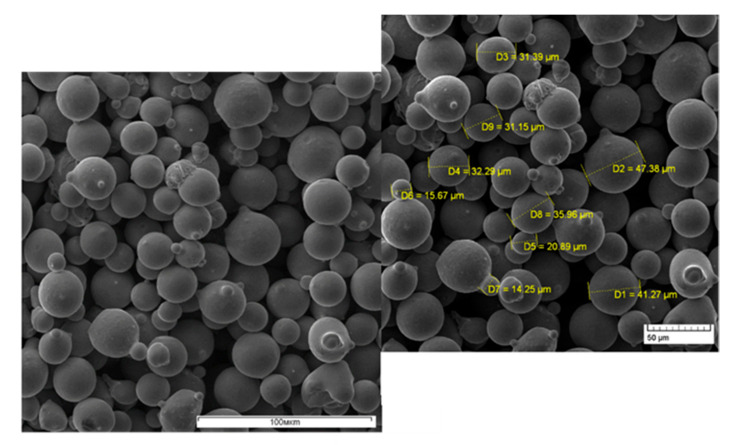
Particles of Ni-Cr-Al-Ti-based superalloy powder.

**Figure 3 materials-16-02088-f003:**
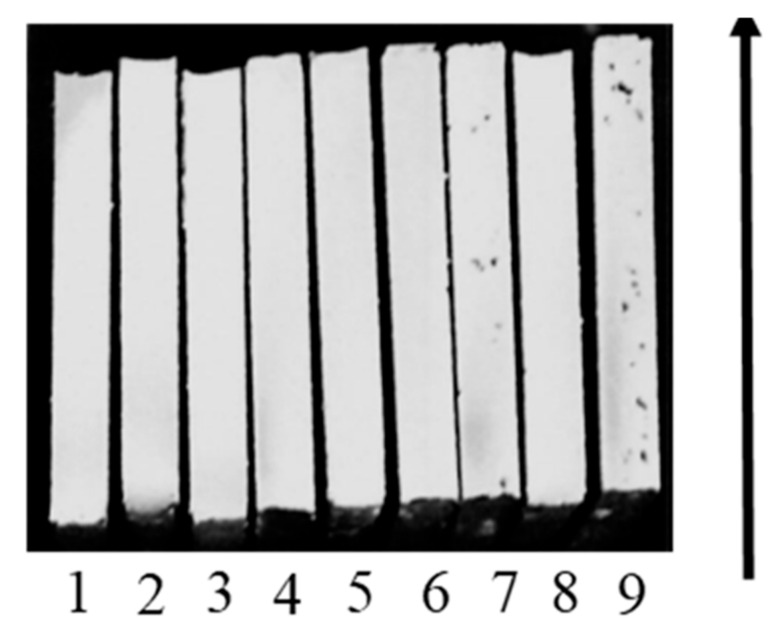
Macrostructure of samples.

**Figure 4 materials-16-02088-f004:**
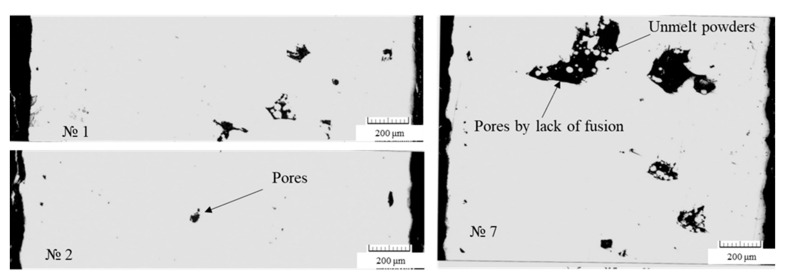
Defects in samples.

**Figure 5 materials-16-02088-f005:**
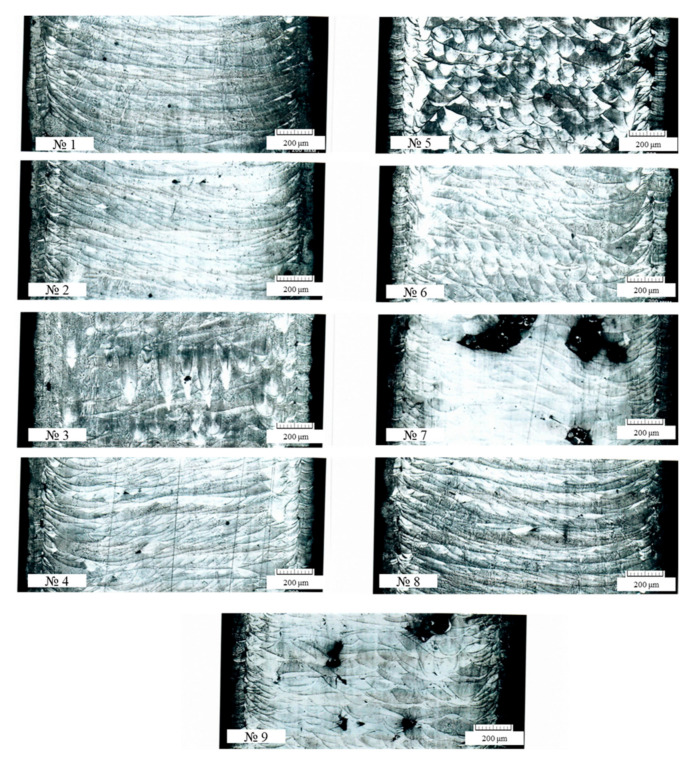
Cross-section comparison of 9 different types of melt pool geometries observed.

**Figure 6 materials-16-02088-f006:**
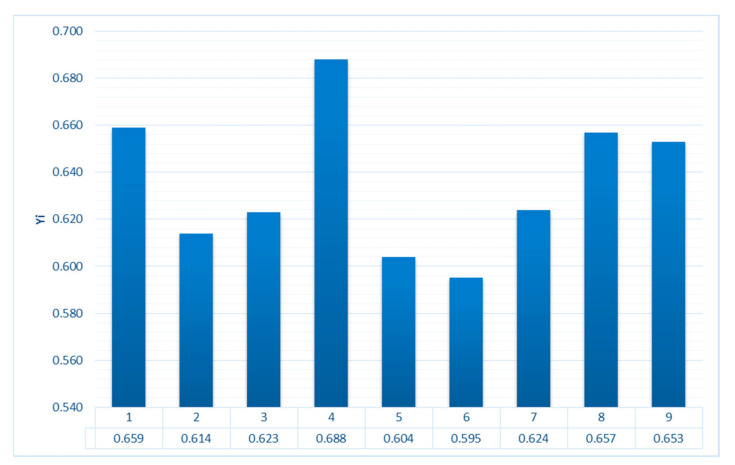
The resulting relational quality scores.

**Table 1 materials-16-02088-t001:** Constants for conducting gray relational analysis.

Yield Stress, σ_0.2_, MPa	Tensile Strength, σ_Β_, MPa	Relative Elongation, δ, %	Microstructure Desirability Rank	Non-Melting Area, mm^2^	Pore Diameter, mm
ξ=0.5	ξ=0.7	ξ=0.5	ξ=0.5	ξ=0.5	ξ=0.5
$x_{i}^{0}=1.2$	$x_{i}^{0}=1.2$	$x_{i}^{0}=1.2$	$x_{i}^{0}=1$	$x_{i}^{0}=1$	$x_{i}^{0}=1$

**Table 2 materials-16-02088-t002:** The experimental plan.

Experiment Number	VED, J/mm^3^	P, W	V, mm/c	h, mm	t, mm
1	66.7	300	600	0.15	0.05
2	37.0	250	900	0.15	0.05
3	97.2	350	600	0.12	0.05
4	34.7	250	1200	0.12	0.05
5	92.6	250	600	0.09	0.05
6	86.4	350	900	0.09	0.05
7	55.6	300	1200	0.09	0.05
8	55.6	300	900	0.12	0.05
9	38.9	350	1200	0.15	0.05

**Table 3 materials-16-02088-t003:** Chemical composition of Ni-Cr-Al-Ti superalloy powder.

Spectrum	Al	Ti	Cr	Co	Ni	Mo	W	Nb	Total
Measured val.	6.08 ± 0.61	3.47 ± 0.12	20.56 ± 0.13	10.43 ± 0.14	54.79 ± 0.33	0.56 ± 0.34	4.11 ± 0.29	-	100.00
TU 1809104096	2.1	3.1	20.0	10.3	base	0.3	3	0.15	100.00
	–	–	–	–		–	–	–	
	2.9	3.9	21.8	12.0		0.9	4	0.35	

**Table 4 materials-16-02088-t004:** Rank assessment of a qualitative characteristic.

Structure Characteristic	Quantitative/Qualitative Characteristic		Desirability Rank “Less Is Better”
No penetration between layers	Significant areas of non-penetration	None	1–2
		observed	9–10
Pores	Single large pores more than 20 microns	Less than 3 in 500 mm^2^	6–8
		More than 3 inclusive in 500 mm^2^	8–10
Structure type	Formation of a molten pool of the “ripple” type—the ratio length/height is not more than 1.8		1–4
	Elongated grains—length/height ratio more than 1.8		5–8

**Table 5 materials-16-02088-t005:** Experimental results.

Experiment Number	Yield Stress σ_0.2_, MPa	Tensile Strength σ_Β_, MPa	Relative Elongation δ, %	Microstructure Desirability Rank	Non-Melting Area, mm^2^		Pore Diameter, mm
					In the Central Part	Along the Edges	In the Central Part
1	797.6	949.0	8.4	6	0.025	0.008	0.025
2	699.5	774.7	5.2	6	0.002	0.00125	0.025
3	680.8	835.5	7.2	8	0.0014	0.0027	0.03
4	709.9	908.4	10.2	6	0.0008	0.0032	0.023
5	704.0	843.0	7.5	7	0.0006	0.00405	0.03
6	699.1	874.1	8.4	7	0.008	0.01008	0.024
7	680.8	808.3	6.7	10	0.18	0.0021	0.022
8	715.5	880.3	9.1	6	0.0006	0.005	0.02
9	668.4	780.5	5.8	10	0.132	0.0035	0.015

**Table 6 materials-16-02088-t006:** Parameter correlation matrix.

Factors/ Experiment Response Characteristics	Yield Stress, σ_0.2_, MPa	Tensile Strength, σ_Β_, MPa	Relative Elongation, δ, %	Microstructure Desirability Rank	Non-Melting Area, mm^2^		Pore Diameter, mm
					In the Central Part	Along the Edges	In the Central Part
VED, J/mm^3^	0.03822	0.19419	0.10204	0.46347	−0.31511	0.40573	**0.71002**
P, W	−0.20763	−0.08614	−0.11456	0.69631	0.29446	0.39081	−0.27730
V, mm/c	−0.39334	−0.31167	−0.03730	−0.43519	0.60847	−0.29888	−**0.77029**
h, mm	0.25979	−0.05090	−0.26377	0.26111	−0.06161	−0.17481	−0.33893

**Table 7 materials-16-02088-t007:** Normalized values of the response characteristics of the experiment.

Experiment Number	Yield Stress, σ_0.2_, MPa	Tensile Strength, σ_Β_, MPa	Relative Elongation, δ, %	Microstructure Desirability Rank	Non-Melting Area, mm^2^		Pore Diameter, mm
					In the Central Part	Along Edges	
1	1.000	1.000	0.640	0.000	0.864	0.236	0.333
2	0.241	0.000	0.000	0.000	0.992	1.000	0.333
3	0.096	0.349	0.407	0.500	0.996	0.836	0.000
4	0.322	0.767	1.000	0.000	0.999	0.779	0.467
5	0.276	0.392	0.467	0.250	1.000	0.683	0.000
6	0.238	0.570	0.653	0.250	0.959	0.000	0.400
7	0.096	0.193	0.300	1.000	0.000	0.904	0.533
8	0.365	0.606	0.793	0.000	1.000	0.575	0.667
9	0.000	0.034	0.120	1.000	0.268	0.745	1.000

**Table 8 materials-16-02088-t008:** The values of the gray relational coefficient $\xi_{ij}$.

Experiment Number	Yield Stress, σ_0.2_, MPa	Tensile Strength, σ_Β_, MPa	Relative Elongation, δ, %	Microstructure Desirability Rank	Non-Melting Area, mm^2^		Pore Diameter, mm
					In the Central Part	Along Edges	
1	0.100	0.140	0.280	0.500	0.068	0.382	0.333
2	0.480	0.840	0.600	0.500	0.004	0.000	0.333
3	0.552	0.596	0.397	0.250	0.002	0.082	0.500
4	0.439	0.303	0.100	0.500	0.001	0.110	0.267
5	0.462	0.566	0.367	0.375	0.000	0.159	0.500
6	0.481	0.441	0.273	0.375	0.021	0.500	0.300
7	0.552	0.705	0.450	0.000	0.500	0.048	0.233
8	0.418	0.416	0.203	0.500	0.000	0.212	0.167
9	0.600	0.816	0.540	0.000	0.366	0.127	0.000

**Table 9 materials-16-02088-t009:** Mechanical properties of cylindrical samples.

Experiment Number	Yield Stress, σ_0.2_, MPa	Tensile Strength, σ_Β_, MPa	Relative Elongation, δ, %
1	620.3	795.1	19.5
2	625.3	801.4	21
3	618.1	799.7	20.7
Average values	621.2	798.7	20.4
Casting alloy [37]	703.0	768.0	3.3

## Data Availability

Not applicable.

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
