# Peer review of "Multiresponse Optimization of Selective Laser Melting Parameters for the Ni-Cr-Al-Ti-Based Superalloy Using Gray Relational Analysis"

_materials, 2023, doi:10.3390/ma16052088_

Round 1
Reviewer 1 Report
A technique called gray relational analysis has been used for process optimization of the SLM process. The manuscript lacks proper explanations and references to several terminologies and mathematical expressions.
The stress-strain curves of the experimental tensile test of at least one sample of each of the 9 tests should be shown in the manuscript.
Detailed comments:
Line-120 and 121: Section 2.2: Is the scanning step the same as the hatch distance? If so, then the same terminology should be used.
Line 125: Table-2: What is the reason or logic behind using these combinations of scanning parameters P, V, h, and t? Authors should suggest evidence/references or explain why these parameters are selected.
Line 188: Table-4: which desirability rank is for which characteristics, it is hard to understand from this table. The third column, which is desirability rank, what is 1-2 for, what is 9-10 for, and what is 6-8 for? How are these ranks decided?
Line 197: What is Sperman's correlation coefficient?
Line 201: What is Fisher's F-test? what is the coefficient of Fisher's F-test?
Line 203: How to decide the desirability rank?
Line 224: How did you calculate these values to obtain the parameter correlation matrix?
Line 225: Section-3.1.: The title of the subsection cannot be "subsection".
Line 230 and 235: what are the parameters 1-3 and 4,5, and 6 in Table-3 for calculating the normalized values? Is there any reference or reason that can be explained for using such a formula for normalization?
Line 242: Please add references for such a mathematical definitions.
Line 272: What is meant by type VII, sample 4 ?
Overall comments:
Why the density of the printed samples has not been chosen as one of the response characteristics?
The following papers can be cited for better relevance and referencing:
Additive manufacturing of lattice structures for high strength mechanical interlocking of metal and resin during injection molding, Additive Manufacturing, Volume 49, 2022.
A bio-inspired design strategy for easy powder removal in powder-bed based additive manufactured lattice structure, Virtual and Physical Prototyping, 17:3, 468-488.
Author Response
Good day!
Thanks for the review.
We tried to take into account all comments and answer all questions.

Reviewer 2 Report
In this manuscript, the author compares and optimizes the optimal technological parameters for scanning Ni-Cr-Al-Ti superalloy basing on the better mechanical properties and fewer defects. This work is very meaningful for the aviation and rocket and space industry. In addition, the Ni-Cr-Al-Ti based superalloy and SLM process are relatively novel and belong to the frontier fields of scientific research. Overall, this is an interesting paper and this reviewer recommends publishing the paper after the following concerns are addressed.
(1) The article structure does not meet the requirements of research papers, and the conclusion part is missing.
(2) The section "Materials and Methods" contains a large number of test results and analyses. Please be sure to rewrite the section to delete test results and write test methods simply.
(3) Table 3 is of little significance, so it is recommended to delete it and just indicate it in the text.
(4) The description in Figs. 4 and 5 is too little. Please add microstructure characteristic analysis and explanation of the variation in the amount of unmelt powder.
(5) Please use quantitative descriptions in the abstract and conclusion, and reinforce the interpretation of the significance of the results.
(6) To confirm the results obtained by the method, add uniaxial stretching results for the worst parameters(sample 6) to compare with the optimal group.
Author Response
Уважаемый рецензент, добрый день!
Спасибо за обзор статьи.
Выкладываю исправленную версию.

Reviewer 3 Report
The paper needs the following improvement:
1. Please check the in-text citations and avoid the citations of too many references at once.
2. The literature study of the paper should be expanded with a state-of-the-art discussion.
3. Please mention the difference in your work from the literature study. In addition, the novelty of this study should be highlighted.
4. Can the authors explain the possible reason for selecting the specific input parameters, i.e., Laser 120 radiation power 250…350 W, scanning speed 600…1200 mm/s, scanning step 0.09…0.15 121 mm?
5. Was there any ASTM standard applied for the mechanical tests?
6. The structure of the paper should be checked: 3.1 subsection?
7. There should be conclusions to the paper. Also, add a discussion section with the results.
8. The discussion section should be explained with justification—agreement and disagreement.
9. Can the authors explain if validation experiments are required to confirm the optimized parameters?
Author Response
Dear Reviewer,
Thank you very much for the review. In the new version of the article, I tried to take into account all your suggestions.

Round 2
Reviewer 1 Report
Agreed with the authors replies.
Author Response
We appreciate the time and effort that the reviewers dedicated to providing feedback on our manuscript and are grateful for the insightful comments on and valuable improvements to our paper.
Reviewer 3 Report
Line 54: Can the authors explain where is the weldability diagram?
Line 160 and 161: Please correct the ranges. It should be laser power (250, 300 and 350) W. Please correct the remaining ranges as well.
Comment #2 (state-of-the-art literature study) and comment # 8 (The discussion section should be explained with justification—agreement and disagreement) have not been addressed.
Round 3
Reviewer 3 Report
Please include weldability diagram in the paper as Figure-1 and cite it with appropriate reference. Thanks
Author Response
Thank you for giving us the opportunity to submit a revised draft of the article. We appreciate the time and effort that you and the reviewers dedicated to providing feedback on our manuscript and are grateful for the insightful comments on and valuable improvements to our paper. We have included a weldability diagram in the article with an appropriate reference.
Kind regards,
authors